# Efficacy and Safety of Two Salts of Trientine in the Treatment of Wilson’s Disease

**DOI:** 10.3390/jcm11143975

**Published:** 2022-07-08

**Authors:** France Woimant, Dominique Debray, Erwan Morvan, Mickael Alexandre Obadia, Aurélia Poujois

**Affiliations:** 1Department of Neurology, Lariboisière Hospital, AP-HP, 75010 Paris, France; france.woimant@live.fr; 2Department of Liver Pediatrics, Necker Hospital, AP-HP, 75015 Paris, France; dominique.debray@aphp.fr; 3Department of Neurology, Rothschild Foundation Hospital, 75019 Paris, France; emorvan@for.paris; 4National Reference Centre for Wilson’s Disease, Rothschild Foundation Hospital, 75019 Paris, France; apoujois@for.paris

**Keywords:** Wilson’s disease, medication adherence, chronic disease, trientine salts, D-penicillamine, zinc salts, efficacy, safety

## Abstract

Background: Wilson’s disease (WD) is one of the few genetic disorders that can be successfully treated with pharmacological agents. Copper-chelating agents (D-penicillamine and Trientine salts) and zinc salts have been demonstrated to be effective. There are two salts of trientine. Trientine dihydrochloride salt (TETA 2HCL) is unstable at room temperature and requires storage at 2–8 °C. Trientine tetrahydrochloride (TETA 4HCL) is a more stable salt of trientine that can be stored at room temperature. No comparative study between both of the salts of trientine has been performed to date. As the two chemical forms were available in France between 1970 and 2009, we conducted a study to evaluate their efficacy and safety profiles. Methods: This retrospective cohort study was conducted by reviewing data from the national WD registry in France. Forty-three WD patients who received TETA 2HCL or TETA 4HCL monotherapy for at least one year until 2010 were included. The primary endpoints were hepatic and neurological outcomes. Secondary endpoints were the events leading to a discontinuation of medication. Results: Changes in medication were common, leading to the analysis of 57 treatment sequences of TETA 4HCL or TETA 2HCL. The mean duration of treatment sequence was significantly longer in the TETA 4 HCL group (12.6 years) than in the TETA 2HCL group (7.6 years) (*p* = 0.011). Ten patients experienced both trientine salts: eight stopped TETA 4 HCL (six had a hepatologic phenotype and two had a neurological phenotype) because this treatment was not available anymore (mean duration 7.4 years). Three of these patients already experienced TETA 2 HCL before the sequence. Two patients with a hepatologic phenotype (one had a previous sequence of TETA 4 HCL before) stopped TETA 2 HCL because of cold storage issues (mean duration 42.8 years). The total number of sequences was 57. All of the patients were clinically stable. No difference in efficacy was detected. Both treatments were well tolerated, except for a case of recurrence of lupus erythematosus-like syndrome in the TETA 2HCL group. The major reason for interruption of TETA 4HCL was due to a discontinuation in production of this salt. The reasons for stopping TETA 2HCL were mainly due to adherence issues largely attributed to the cold storage requirement. Conclusions: The two salts of trientine were effective in treating patients with WD. However, interruption of TETA 2HCL was frequent, linked to the cold storage requirement. As adherence to treatment is a key factor in the successful management of WD, physicians need to be even more vigilant in detecting adherence difficulties in patients receiving treatment with TETA 2HCL.

## 1. Introduction

Wilson’s disease (WD) is an autosomal recessive disorder characterized by pathological copper accumulation in many organs, initially the liver, and then essentially the cornea and brain. It is caused by homozygous or compound heterozygous mutations in the ATP7B gene which encodes a transmembrane copper-transporting P-type ATPase [1]. WD is one of the few genetic disorders that can be successfully treated with pharmacological agents. The treatment is based on the generation of a negative copper balance. Copper-chelating agents and zinc salts have been demonstrated to be effective in the treatment of WD, associated with a low copper diet [2]. However, the best therapeutic approach remains controversial because no randomized controlled trials have compared these treatments and the use of drugs depends mainly on center experience and access to treatment in different countries or regions. Treatment is a lifelong necessity and should be started as early as possible. Whatever the chosen medical therapy, non-adherence to or discontinuation of therapy is associated with a high risk of very severe hepatic or neurologic deterioration [3]. The optimum goal for the patients requiring this lifelong medical therapy should therefore be to limit the side effects and difficulties associated with treatment dispensation and conservation.

Copper-chelating agents (D-penicillamine and Trientine salts) bind with excess copper, forming a stable complex which is excreted mainly in the urine. It has been suggested that trientine salts may also decrease intestinal copper absorption [2,4]. Zinc salts decrease the intestinal absorption of copper, inducing the synthesis of metallothioneins, proteins that sequester copper in the enterocytes [5]. Zinc salts are indicated in pre-symptomatic patients and during the maintenance phase of treatment [2], but some data indicate that zinc may also be considered in patients who exhibit neurological symptoms during the acute phase of the disease [6]. Zinc is generally well-tolerated in adults, although gastritis and nausea may lead to discontinuation of the treatment [7]. In pre-symptomatic children, gastrointestinal adverse effects are present in nearly 20% of the patients, associated with poor efficacy [8].

D-penicillamine is the reference treatment in many European countries, but severe adverse effects are frequent, leading to a discontinuation of this therapy in up to 30 % of patients [9]. The other copper chelators are trientine salts, currently indicated in WD patients who are intolerant to D-penicillamine. There are two currently available trientine salts. The trientine dihydrochloride salt (TETA 2HCL) is unstable at room temperature and requires storage between 2 and 8 °C. In Europe, TETA 2HCL was approved by the Medicines Health and Regulatory Agency (MHRA) in the United Kingdom (UK) for the treatment of WD in 1985, and was supplied to some other European Union (EU) countries. In France, TETA 2HCL (Trientine^®^ from Univar, Downers Grove, IL, USA) was used through a compassionate use program for those patients intolerant to D-penicillamine and was only dispensed by hospital pharmacies. This salt is now marketed in the EU as Cufence^®^, following EU marketing authorization in 2019. Trientine tetrahydrochloride (TETA 4HCL) is a more stable salt of trientine that can be stored at room temperature. In France, TETA 4HCL was available from the mid-1970s until 2009, as a hospital preparation supplied by AGEPS (Agence Générale des Equipements des Produits de Santé) of the Assistance Publique—Hôpitaux de Paris. This salt was granted a European marketing authorization in 2018 and is currently marketed in Europe as Cuprior^®^ (Orphalan, Paris, France).

No comparative study between both of the salts of trientine (dihydrochloride and tetrahydrochloride) has previously been performed. As the two chemical forms were available in France between 1970 and 2009, we conducted a study to evaluate the efficacy and safety profiles of both of the salts.

## 2. Materials and Methods

### 2.1. Ethics Approval and Consent to Participate

This study was approved by the Institutional Review Board of HUPNVS, Paris 7 University, AP-HP (n°1343579). An informed consent was obtained from all of the subjects and/or their legal guardian(s). All of the patients signed a written consent form. All of the methods were carried out in accordance with relevant guidelines and regulations, and in accordance with the Declaration of Helsinki.

### 2.2. Patients

This retrospective cohort study was conducted by reviewing data from the national WD registry. The WD patients who were followed in Lariboisière hospital—Paris (National Centre for Wilson’s disease) and treated until 2010 were selected. This final date corresponded to the year after which the production of TETA 4HCL by AGEPS was discontinued. The diagnosis of WD was based on clinical symptoms, abnormal copper metabolism and genetic testing with a Leipzig score ≥4 [2]. Only the patients receiving TETA 2HCL or TETA 4 HCL monotherapy for at least one year were included. The specific duration of one year of trientine monotherapy was considered as the minimum time required to show a treatment effect. The patients were in the initial or maintenance phase of treatment and received trientine as a first, second or third line treatment. The patients who successively received both forms of trientine (TETA 4HCL and 2HCL) were included in the analysis. The patients who received either TETA 2HCL or TETA 4HCL in association with zinc salts were excluded.

### 2.3. Analysis of Treatments

The different courses of treatment with TETA 4HCL and 2HCL were identified and the treatment sequences of TETA 4HCL or 2 HCL with a follow-up period superior to one year of continuous treatment were analyzed. The events leading to a discontinuation of medication were recorded and classified.

### 2.4. Baseline Comparison of Treatments

The clinical and laboratory data were recorded at the beginning and at the end of each trientine treatment sequence; the duration of the sequence was noted. The patients included in the study were divided into two groups, according to the absence or presence of neurological symptoms. The hepatic assessment included clinical symptoms, measurement of serum transaminase levels, bilirubin and prothrombin time (PT). The presence of cirrhosis (typical findings on imaging and/or presence of clinical signs of portal hypertension) was recorded. The neurological evaluation was based on clinical symptoms. The presence of Kayser–Fleischer rings at slit-lamp examination was documented. The adherence to treatment was recorded by reports of compliance from the patients that were recorded in their medical records.

### 2.5. Study Endpoints

Primary endpoints were hepatic and neurological outcomes. The hepatic outcome was based on clinical symptoms and a course of liver enzymes and liver function tests. The neurological outcome was evaluated by neurological symptoms. Both hepatic and neurologic outcomes were scored as follows: unchanged, improved or deteriorated. The evolution of Kayser–Fleischer rings was also scored as unchanged, improved, disappeared or increased.

The secondary endpoints were the events leading to a discontinuation of medication. The reasons for treatment interruption or discontinuation were classified: loss of efficacy, adverse events, treatment non-adherence, manufacturing interruption.

### 2.6. Statistical Analyses

The quantitative variables were expressed as median (interquartile range) and the categorical variables as frequencies and percentages. Comparisons between two groups were made using the Student U test for continuous variables and the Fisher exact test for qualitative variables if the frequency was <5, otherwise the chi-squared test was used.

## 3. Results

### 3.1. Study Group

From the 248 WD patients recorded in the national registry and who were followed at the Lariboisière hospital before 2010, 62 received at least one sequence of treatment with a trientine salt. Nineteen patients were excluded because they received a zinc salt in combination with trientine over the treatment period. Thus, 43 patients were included in the study (Figure 1). Twenty-three were male (53.5%) and the age at diagnosis was 21 ± 9.3 years (min 5.6; max 46.3). Nine patients (20.9%) were diagnosed at the pre-symptomatic stage via familial screening, 19 presented with hepatic symptoms and 15 with neurological symptoms. Sixteen patients (37 %) had cirrhosis. Trientine was the first-line treatment for four patients (9.52%). Trientine was prescribed as a second-line treatment after D-penicillamine in 35 patients, zinc salts in 2 patients and D-penicillamine associated with zinc salts in 2 other patients. D-penicillamine was stopped due to WD aggravation (two cases) or due to the occurrence of adverse events in the remaining patients. The most common adverse events were renal disorders, thrombocytopenia and neutropenia, skin rash, digestive disorders and, less frequently, arthralgia, myasthenia-like syndrome and lupus-like syndrome. Zinc salts were stopped due to gastric irritation and, in one case, due to an increase in liver enzymes.

### 3.2. Treatment Sequences

Changes in the trientine treatment were common in this cohort. The 43 patients received 57 trientine monotherapy treatment sequences. This included 10 patients who received both TETA 4HCL and TETA 2HCL in different sequences (with a duration of more than one year); 2 patients who received only TETA 4HCl and 31 patients who received only TETA 2HCL. This corresponded to 57 trientine treatment sequences: 13 sequences with TETA 4HCL and 44 sequences with TETA 2HCL (Figure 1).

### 3.3. Baseline Characteristics of TETA Treatment Sequences

The mean sequence duration was significantly longer in the TETA 4HCL group, 151.7 ± 111.1 months, vs 91.1 ± 58.1 months in the TETA 2HCL group (*p* = 0.011).

Table 1 presents a comparison of the baseline parameters at the initiation of the treatment sequence in the two groups, TETA 2HCL and TETA 4HCL. The laboratory analyses were available only for a subset of the patients due to the retrospective nature of the study. However, there were no statistically significant differences between the groups relating to sex, age, laboratory values and delay in onset of treatment. Regarding the initial phenotype, more of the patients in the TETA 4HCL group had neurological signs: 62% versus 43% in the TETA 2HCL group.

### 3.4. Patient Outcomes

The analysis of the evolution of hepatic and neurologic outcomes shows that the majority of the patients either improved clinically or their symptoms stabilized under the TETA treatment sequences (Table 2). The parameters relative to hepatic function, in particular serum transaminases, tended to improve (13 (29.55%)) in the TETA 2HCL group vs. 3 (23.08%) in the TETA 4HCL group), with no statistically significant differences observed between the two groups. Nevertheless, three patients worsened in the TETA 2HCL group.

Table 3 details the evolution between the two subgroups, based on the presence of neurological symptoms at the sequence initiation. When the hepatic symptoms were isolated at the initiation of the trientine sequence, no neurological symptoms developed. When only neurological symptoms were evident at the initiation of the treatment sequence, they improved or remained unchanged for all except for one TETA 2HCL sequence.

Kayser–Fleischer (KF) ring evolution was comparable in both of the treatment groups. In one sequence of the TETA 4HCL group, a slight increase in the ring was reported, without a deterioration in neurological and hepatic disease (Table 4).

### 3.5. Reasons for Discontinuation of Trientine Treatment and Adverse Effects

No adverse effects were observed during the TETA 4HCL sequences (mean duration: 12.6 years). All 13 of the TETA 4HCL sequences were stopped during the study period: 11 (85%) due to the fact that manufacturing of the hospital preparation was discontinued, 1 due to difficulties with supply and 1 due to an increase in the Kayser–Fleischer ring, without neurologic or hepatic deterioration.

The mean duration of treatment sequence was 7.6 years for TETA 2HCL. At the end of the study period, 26 (60%) of the TETA 2HCL monotherapy treatments were still ongoing. In three patients, zinc was added to TETA 2HCL due to hepatic or neurological deterioration in two cases, and as a result of an increase in liver copper without hepatic deterioration in one case. A bad adherence to TETA 2HCL was suspected in these three cases. The reasons for stopping TETA 2HCL were mainly due to adherence to medication issues (11 cases), generally linked to the requirement for cold storage. Other reasons included one case of lupus erythematosus-like syndrome in a patient with a previous diagnosis of lupus erythematosus during treatment with D-penicillamine, and one liver transplantation for suspicion of hepatocellular carcinoma. Two patients died (salivary gland neoplasm and suicide); these deaths were not considered as related to WD.

## 4. Discussion

Now that TETA 4HCL has received a European marketing authorization and is being marketed in Europe (as Cuprior^®^), this study comparing the efficacy and safety of TETA 4HCL to TETA 2HCL in everyday clinical practice is important, since many patients are still taking TETA 2HCL. Efficacy and safety are closely interrelated because any switch in treatment is usually linked to a lack of efficacy, observance issues, adverse effects or difficulties with treatment adherence.

In France, between 1970 and 2009, both of the trientine salts were available. All of the patients with WD in France are included in a national registry, allowing the possibility of conducting this retrospective study. Trientine monotherapy for more than 12 months was evaluated in 43 patients out of the 248 patients included in the WD registry in 2010, which represents a large cohort in this rare disease.

In accordance with French guidelines, trientine was mainly prescribed as a second-line treatment after D-penicillamine (81% of the patients). This switch was mainly due to the occurrence of adverse events. Adverse drug reactions are commonly reported with D-penicillamine treatment and are serious enough to lead to at least 30% of patients on D-penicillamine discontinuing the drug [10]. Four patients received zinc salts as a first-line treatment, associated or not with D-penicillamine. Discontinuation of zinc therapy, due to adverse effects such as gastrointestinal symptoms, is common in patients with WD in children, as in adults [7,8].

The study population is representative of the wide WD population, as it includes children and adult patients. The mean age at diagnosis was 20 years (range 5.6 years to 46.3 years). The majority of the patients presented with liver disease. These data are comparable to those of former studies [7,9]. Although there was non-random allocation to the treatment group, the baseline characteristics of the patients at the beginning of the sequences were relatively balanced.

The majority of the patients either improved or stabilized their symptoms under trientine. No differences in efficacy were detected when assessing the TETA 4HCL and 2HCL treatment sequences for changes in hepatic and neurological symptoms. The parameters relative to hepatic function, in particular serum transaminase levels, tended to improve in both of the groups. Many studies have demonstrated the effectiveness of TETA 2HCL, showing in addition that fewer side effects are observed than with D-penicillamine [9,11]. In this study, three patients on TETA 2HCL had hepatic or neurological deterioration and a bad adherence to treatment was highly suspected in these cases. In WD, during long-term follow-up, the most important cause of hepatic and/or neurologic worsening, leading sometimes to death, is non-adherence to WD treatment [12,13,14,15]. Up to 50% of patients are non-compliant with treatment [16,17]. The identification of the factors which compromise adherence to medication remains difficult, and findings are often contradictory. However, it is evident that the barriers to treatment dispensation or conservation should be minimized. TETA 2HCL requires refrigerated storage between 2 and 8 °C. This is certainly an important disadvantage for patients who need to take the drug several times throughout the day, while studying, working or travelling, on a daily basis for the duration of their lives.

In this study, the main reason for discontinuation of the drug was the production shutdown of one of the biochemical salts, TETA 4HCL. All 13 TETA 4HCL treatments were stopped due to cessation of manufacturing or supply difficulties, except for 1 patient whose Kayser–Fleischer rings increased. No adverse events were reported in this group following an average treatment duration of 12.5 years. In the TETA 2HCL group (*n* = 44), only one adverse event was reported. This was a case of lupus erythematosus-like syndrome in a patient who already had presented with penicillamine-induced lupus erythematosus-like syndrome. However, 11 treatment sequences were stopped as a result of difficulties in treatment adherence due to the refrigeration requirements (often creating problems for patients who needed to work away from home or travel) for TETA 2HCL. The published guidelines indicate that adverse effects are rarely observed with TETA 2HCL treatment: urticaria, reversible anemia and lupus-like reactions are described as the key potential side effects [2,18]. In our study, TETA 4HCL was also shown to be well tolerated, consistent with the review of Allery that describes a TETA 4HCL safety profile comparable to that of TETA 2 HCL [19].

The number of patients treated with trientine as a first-line therapy is small and does not allow for a satisfactory analysis of this sub-group. However, six other patients started trientine therapy early in the course of this lifelong disease, during the first three months following diagnosis. Trientine was introduced in these patients as a result of early adverse events associated with D-penicillamine. The clinical evolution of these patients was not different to that of the whole cohort.

This study has certain limitations, including its retrospective nature and the lack of randomization. The rarity of the disease, the fact that trientine was used in France via a compassionate use program and that manufacturing of TETA 4HCL was discontinued in 2009, explain the low number of patients included in the study. At that time, adherence was subjectively assessed during the medical examination and not with dedicated scores, such as the Morisky score [20]. However, it was possible to analyze data quite exhaustively over long periods of treatment under trientine (7.6 years for TETA 2HCL and 12.6 years for TETA 4HCL).

In conclusion, both of the trientine salts were equally effective in controlling WD. Adverse events were infrequent. In WD, the adherence to medication is a key factor for treatment success. Interruption in the TETA 2HCL therapy was frequent, linked to the requirement for cold storage. The physicians therefore have to be even more vigilant to detect non-adherence to medication as early as possible in the patients being treated with TETA 2HCL.

## Figures and Tables

**Figure 1 jcm-11-03975-f001:**
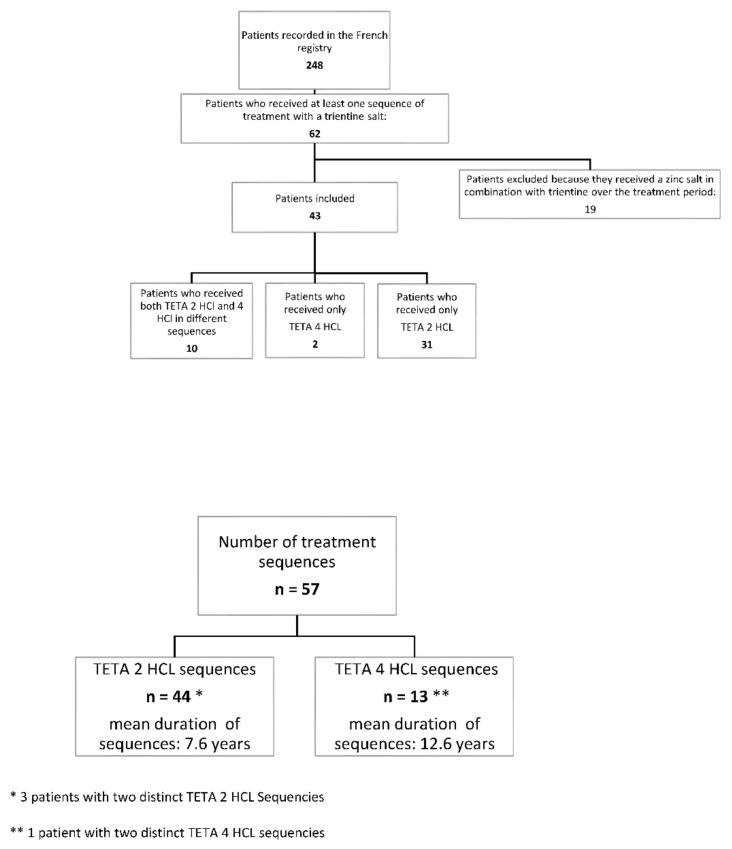
Flow-chart of patients included and treatment sequences.

**Table 1 jcm-11-03975-t001:** Baseline characteristics at Trientine sequence initiation.

	Trientine Treatment Sequence		
	TETA 4HCL (*n =* 13)	TETA 2HCL (*n =* 44)	*p*-Value
**Sex ratio** (male/female)	6/7	20/24	0.965
**Mean age at trientine initiation** (years ± SD)	28.7 ± 9.2	28.8 ± 13.6	0.979
**Clinical form at sequence initiation**			
Hepatic form	5/13 (38.46%)	25/44 (56.82%)	0.407
Neurological form	8/13 (61.54%)	19/44 (43.18%)	0.244
Presence of cirrhosis	4/13 (30.77%)	16/44 (36.36%)	0.710
**First-line treatment**	2/13 (15.38%)	2/44 (4.55%)	0.179
**Delay between sequence initiation and diagnosis** (years) (mean ± SD/ median/min–max)	5.3 ± 6.2/1.6/0.0–20.6	7.7 ± 10.1/3.2/0.0–36.4	0.414
**Duration of treatment** (months) (mean ± SD/median/min–max)	151.7 ± 111.1/138.9/22.8–391.4	91.1 ± 58.1/78.9/12.9–254.3	0.011
**ALT** (IU/l)			0.936
N	9 (69.33%)	32 (72.73%)	
mean ± SD	56.3 ± 43.4	57.7 ± 43.6	
**AST** (IU/l)			0.727
N	9 (69.33%)	32 (72.73%)	
mean ± SD	42.9 ± 31.7	47.6 ± 36.7	
**Total bilirubin** (µmol/L)			0.853
N	4 (30.77%)	21 (47.73%)	
mean ± SD	16.5 ± 10.1	15.3 ± 11.6	
**PT** (% of normal)			0.802
N	5 (38.46%)	25 (56.82%)	
mean ± SD	86 ± 14.3	83.6 ± 20.0	
**Platelets** (/mm^3^)			0.436
N	7 (53.85%)	29 (63.64%)	
mean ± SD	156.9 ± 30.2	182.0 ± 82.2	
**Ceruloplasmin** (g/L)			0.519
N	6 (46.15%)	21 (47.73%)	
mean ± SD	0.04 ± 0.03	0.03 ± 1.1	
**Serum copper** (µmol/L)			0.273
N	8 (61.54%)	22 (50.00%)	
mean ± SD	3.1 ± 2.3	6.0 ± 7.1	
**Urine copper at start of treatment sequence** (µmol/L)			0.848
N	6 (46.15%)	22 (50.00%)	
mean ± SD	5.5 ± 5.6	6.4 ±10.8	
**Urine copper at end of treatment sequence** (µmol/L)			0.623
N	6 (46.15%)	39 (88.64 %)	
mean ± SD	3.9 ± 2.9	5.0 ± 5.6	

**Table 2 jcm-11-03975-t002:** Hepatic and neurologic evolution in all patients.

	Trientine Treatment Sequence		
	TETA 4HCL (*n =* 13)	TETA 2HCL (*n =* 44)	*p*-Value
**Hepatic outcome**			0.842
Improved	3 (23.08%)	13 (29.55%)	
Unchanged	10 (76.92%)	29 (65.91%)	
Worsened	0	2 (4.55%)	
**Neurologic outcome**			0.172
Improved	4 (30.77%)	12 (27.27%)	
Unchanged	9 (69.23%)	31 (70.46 %)	
Worsened	0	1 (2.27%)	
**Mean changes of serum transaminases between the start and end of the sequence**			
ALT (IU/L)	−25.3 ± 35.5	−1.2 ± 44.3	0.164
AST (IU/L)	−10 ± 22.8	−7 ± 31.5	0.802

**Table 3 jcm-11-03975-t003:** Outcome in patients based on presence of neurological symptoms at the sequence initiation.

	Trientine Treatment Sequence	
Patients without Neurological Symptoms	TETA 4HCL (*n =* 5)	TETA 2HCL (*n =* 25)
**Hepatic outcome**		
Improved	3 (60.00%)	11 (44.00%)
Unchanged	2 (40.00%)	12 (48.00%)
Worsened	0 (0.00%)	2 (8.00%)
**Neurological outcome**		
Absent	5 (100.00%)	25 (100.00%)
**Patients with neurological symptoms**	**TETA 4HCL** **(*n =* 8)**	**TETA 2HCL** **(*n =* 19)**
**Hepatic outcome**		
Improved	0	2 (10.53%)
Unchanged	8 (100.00%)	17 (89.47%)
Worsened	0	0
**Neurological outcome**		
Improved	4 (50.00%)	12 (63.16%)
Unchanged	4 (50.00%)	6 (31.58%)
Worsened	0	1 (5.26%)

**Table 4 jcm-11-03975-t004:** Evolution of the Kayser-Fleischer ring between sequence initiation and sequence end.

	Trientine Treatment Sequence	
	TETA 4HCL (*n =* 13)	TETA 2 HCL (*n =* 44)
**Information not available**	1 (7.69 %)	1 (2.27 %)
**Present at treatment sequence initiation**	8 (61.54%)	17 (38.63%)
**At sequence end**		
Increase	1 (12.50%)	0
Decrease	2 (25.00%)	8 (47.06%)
Disappearance	5 (62.50%)	8 (47.06%)
Unchanged	0	1 (5.88%)

## Data Availability

The datasets generated and/or analyzed during the current study are not publicly available due to privacy related to the French Wilson’s Disease National Registry but are available from the corresponding author on reasonable request.

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
