# Peer review of "Efficacy and Safety of Two Salts of Trientine in the Treatment of Wilson’s Disease"

_jcm, 2022, doi:10.3390/jcm11143975_

Round 1

Reviewer 1 Report

Due to the better stability of Trientine tetrahydrochloride (TETA  4HCL) than Trientine dihydrochloride salt (TETA 2HCL) at room temperature, it is important to compare their efficacy and safety before TETA 4HCL being used widely. This retrospective research has supplied the preliminary data and deserved to be published. However, it should be minor revised for the following reasons.

The sample size of patients with TETA 4 HCl treatment is small, of which, 10 Patients had received both TETA 2 HCl and 4 HCl in different sequences, and only 2 Patients received TETA 4 HCL monotherapy. The author used the number of treatment sequences to analyze the efficacy of TETA 4 HCl, which means that its efficacy is likely to be mixed and confused with that of TETA 2HCl if the later been used before. Please add the detailed drug usage information, such as the duration, the disease status at the time point of drug shifting in the 10 patients who had received both TETA 2 HCl and 4 HCl treatment.

Similarly, for the TETA 2 HCl treatment group,10 Patients received both TETA 2 HCl and 4 HCl,31 Patients received only TETA 2 HCL.  The total patients number is 41. However, the sample size of TETA 2 HCL treatment sequences is 44. Please also add the detailed information of the drug usage.

Author Response

Dear reviewer,

Thank you very much for your kind attention and for this review with very constructive remarks.

We would like to submit the revised version of our article entitled “Efficacy and safety of two salts of trientine in the treatment of Wilson disease” by France Woimant, Dominique Debray, Erwan Morvan, Mickael Alexandre Obadia and Aurélia Poujois.

Please find attached our responses.

Due to the better stability of Trientine tetrahydrochloride (TETA  4HCL) than Trientine dihydrochloride salt (TETA 2HCL) at room temperature, it is important to compare their efficacy and safety before TETA 4HCL being used widely. This retrospective research has supplied the preliminary data. However, it should be minor revised for the following reasons. 

The sample size of patients with TETA 4 HCl treatment is small, of which, 10 Patients had received both TETA 2 HCl and 4 HCl in different sequences, and only 2 Patients received TETA 4 HCL monotherapy.

> Thank you for your comment. Only 2 Patients received TETA 4 HCL monotherapy as first line therapy because, in accordance with the French guidelines, prescription of D-penicillamine is recommended as first line and the prescription of trientine salts comes in second line treatment. Moreover, TETA 4 HCL was less prescribed compared TETA 2 HCL because It was only available as a hospital preparation in Paris, with supply difficulties for patients living far from Paris.

The author used the number of treatment sequences to analyze the efficacy of TETA 4 HCl, which means that its efficacy is likely to be mixed and confused with that of TETA 2HCl if the later been used before. Please add the detailed drug usage information, such as the duration, the disease status at the time point of drug shifting in the 10 patients who had received both TETA 2 HCl and 4 HCl treatment. Similarly, for the TETA 2 HCl treatment group,10 Patients received both TETA 2 HCl and 4 HCl,31 Patients received only TETA 2 HCL.  The total patient’s number is 41. However, the sample size of TETA 2 HCL treatment sequences is 44. Please also add the detailed information of the drug usage.

> Thank you very much for your attention. As mentioned in the flow chart we included 43 patients. 2 patients received TETA 4HCL only, 31 received TETA 2HCL only and 10 patients received both 2HCL and 4HCL. Among them, 6 had a hepatologic phenotype and 2 had a neurological phenotype, with a mean duration of 7.4 years. The 2 other patients with a hepatologic phenotype stopped TETA 2 HCL because of cold storage issues (mean duration 42.8 years). Among those 10 patients, 3 patients had each 3 sequences (TETA 2HCL=> TETA 4HCL=>TETA 2HCL) because TETA 4HCL was no more commercialized, so they moved back to TETA 2HCL. They were clinically stable. One patient who was clinically stabilized on TETA 2HCL switched back to TETA 4HCL because she had issues of cold storage. This explains why there are 13 sequences of TETA 4HCL for 12 patients and 44 sequences for TETA 2HCL for 41 patients. We have added a text to explain this in our article (page 10, line 214) and a line in the figure 1 to make this clearer.

Reviewer 2 Report

Wilson’s disease (WD) is an autosomal recessive genetic disorder of copper metabolism leading to liver or brain injury due to accumulation of copper. Pharmacological therapy comprises chelating agents (penicillamine, trientine) and zinc salts which seem to be very effective. This study is a retrospective analysis in a group of 43 patients with WD receiving two forms of trientine  dihydrochloride salt the first TETA 2HCL which is unstable at room temperature and a second, the trientine tetrahydrochloride (TETA 4HCL) which is more stable.  These two chelating agents were not always available over time, and a non-uniform compliance of drugs in patients with neurological and psychiatric disorders has sometimes been a cause of concern.

Response to treatment is usually assessed based on clinical evaluation of neurologic disorders, normal liver function tests and monitoring of copper metabolism markers.

Due to the retrospective nature of the paper some major biases exist. These are also secondary to the difficulty of tracking adherence to therapy and the impossibility of assessing the neurological status with defined clinical scoring systems. These points should be better stressed in the discussion.

I have 3 questions about this cohort pf patients:

    1. The long-term survival in WD patients seems to be very similar as for the general population when the disease is early diagnosed and correctly treated. WD patients with a longer delay from diagnosis to therapy and who present with neurological and psychiatric symptoms have worse quality of life. My first question is: can you specify how many patients presented specific psychiatric disturbances between the two groups; adolescents and patients with psychiatric disorders usually have more problems with adherence to treatment.

     2. It is usually advised to patients with WD to avoid copper-rich food products (e.g. shellfish, nuts, chocolate, mushrooms) as long as liver tests remain increased. My second question is: can you specify how many of the 43 patients were put in a specific diet avoiding copper-rich food and if specific diet recommendation were done along the TETA treatment.

    3. Liver transplantation (LT) has a well-defined role in Wilsonian acute hepatic failure (acute on chronic liver failure) according to the prognostic score. My third question is: how many patients received a LT in this cohort or at least how many patients were evaluated or enlisted for LT?

Author Response

Dear reviewer,

Thank you very much for your kind attention and for this review with very constructive remarks.

We would like to submit the revised version of our article entitled “Efficacy and safety of two salts of trientine in the treatment of Wilson disease” by France Woimant, Dominique Debray, Erwan Morvan, Mickael Alexandre Obadia and Aurélia Poujois.

Please find attached our responses.

Wilson’s disease (WD) is an autosomal recessive genetic disorder of copper metabolism leading to liver or brain injury due to accumulation of copper. Pharmacological therapy comprises chelating agents (penicillamine, trientine) and zinc salts which seem to be very effective. This study is a retrospective analysis in a group of 43 patients with WD receiving two forms of trientine dihydrochloride salt the first TETA 2HCL which is unstable at room temperature and a second, the trientine tetrahydrochloride (TETA 4HCL) which is more stable. These two chelating agents were not always available over time, and a non-uniform compliance of drugs in patients with neurological and psychiatric disorders has sometimes been a cause of concern.

Response to treatment is usually assessed based on clinical evaluation of neurologic disorders, normal liver function tests and monitoring of copper metabolism markers.

Due to the retrospective nature of the paper some major biases exist. These are also secondary to the difficulty of tracking adherence to therapy and the impossibility of assessing the neurological status with defined clinical scoring systems. These points should be better stressed in the discussion.

> Indeed, the retrospective design introduces some major bias. We have added a sentence (page 14, line 335) in the discussion to point the lack of precise neurological evaluation with a validated scale such as UWDRS. I hope this will make this bias clearer to the reader.

I have 3 questions about this cohort of patients:

  1. The long-term survival in WD patients seems to be very similar as for the general population when the disease is early diagnosed and correctly treated. WD patients with a longer delay from diagnosis to therapy and who present with neurological and psychiatric symptoms have worse quality of life. My first question is: can you specify how many patients presented specific psychiatric disturbances between the two groups; adolescents and patients with psychiatric disorders usually have more problems with adherence to treatment.

> Thank you very much for this very interesting question. Indeed, a wide range of psychiatric symptoms can be associated with Wilson disease. In this retrospective study, a lack of clinical data can’t allow us to answer to this question. Indeed, a significant proportion of these patients can have psychiatric symptoms such as depression or cognitive disorders in their life. But this lack of data shouldn’t significantly modify the adherence issues reported in our article. This is particularly well explained in the article of Jacquelet et al. published in Journal of Inheritance Metabolic Disease in November 2021, showing there is no significant correlation between depression and adherence to treatment.

Reference: Jacquelet E, Poujois A, Pheulpin MC, Demain A, Tinant N, Gastellier N, Woimant F. Adherence to treatment, a challenge even in treatable metabolic rare diseases: A cross sectional study of Wilson's disease. J Inherit Metab Dis. 2021 Nov;44(6):1481-1488. doi: 10.1002/jimd.12430. Epub 2021 Sep 21. PMID: 34480375.

  1. It is usually advised to patients with WD to avoid copper-rich food products (g. shellfish, nuts, chocolate, mushrooms) as long as liver tests remain increased. My second question is: can you specify how many of the 43 patients were put in a specific diet avoiding copper-rich food and if specific diet recommendation were done along the TETA treatment.

> It is correct, all the patients of our cohort were advised to have a copper-poor regimen during the first years of chelating treatment, at least until their liver tests got normalized, in accordance with the international guidelines. Please notice that nearly all patients kept this copper-poor regimen food diet during all the study, mainly because they got used to it for ages and they regularly see the dietician. The precise quantification of copper daily dose wasn’t available due to the retrospective acquisition of data.

  1. Liver transplantation (LT) has a well-defined role in Wilsonian acute hepatic failure (acute on chronic liver failure) according to the prognostic score. My third question is: how many patients received a LT in this cohort or at least how many patients were evaluated or enlisted for LT

> One patient received a LT in our cohort, and this patient stopped the chelating treatment after his transplantation. No other patients were evaluated or enlisted for LT to our knowledge.

Once again thank you very much for your kind attention. We stay available if needed.

Round 2

Reviewer 2 Report

All issues raised by the reviewer have been satisfactorily addressed by the authors.